# Sheng Xue Ning as a Novel Agent that Promotes SCF-Driven Hematopoietic Stem/Progenitor Cell Proliferation to Promote Erythropoiesis

**DOI:** 10.3390/biom14091147

**Published:** 2024-09-11

**Authors:** Yueying Zeng, Chunlu Li, Fei Yang, Ling Zhang, Wanqi Xu, Long Wang, Anguo Wu, Wenjun Zou, Jianming Wu, Feihong Huang

**Affiliations:** 1Luzhou Key Laboratory of Activity Screening and Druggability Evaluation for Chinese Materia Medica, School of Pharmacy, Southwest Medical University, Luzhou 646000, China; 20210599120052@stu.swmu.edu.cn (Y.Z.); yf1564802295@gmail.com (F.Y.); 20220599120033@stu.swmu.edu.cn (L.Z.); 20230599120053@stu.swmu.edu.cn (W.X.); wanglong1226@swmu.edu.cn (L.W.); wuanguo@swmu.edu.cn (A.W.); 2School of Pharmacy, Chengdu University of Traditional Chinese Medicine, Chengdu 611137, China; lichunlu49@gmail.com (C.L.); zouwenjun@cdutcm.edu.cn (W.Z.); 3School of Basic Medical Sciences, Southwest Medical University, Luzhou 646000, China

**Keywords:** Sheng Xue Ning, erythropoiesis, hematopoietic stem/progenitor cell, mesenchymal stem cell, hematopoietic cytokine, stem cell factor

## Abstract

Stimulating erythropoiesis is essential in the treatment of various types of anemia. Sheng Xue Ning (SXN) is commonly used in China as an iron supplement to treat iron deficiency anemia, renal anemia, and anemia in pregnancy. This research reports a novel effect of SXN in enhancing the proliferation of hematopoietic stem/progenitor cell (HSPC) to promote erythropoiesis in the bone marrow, which is distinct from conventional iron supplements that primarily aid in the maturation of red blood cells. Employing a model of hematopoietic dysfunction induced by X-ray exposure, we evaluated the efficacy of SXN in restoring hematopoietic function. SXN significantly promoted the recovery of peripheral erythroid cells and enhanced the proliferation and differentiation of Lin^−^/c-KIT^+^/Sca-1^+^ HSPC in mice exposed to X-ray irradiation. Our results showed that SXN elevated the expression of stem cell factor (SCF) and activated the SCF/c-KIT/PI3K/AKT signaling pathway, facilitating the proliferation and differentiation of HSPC. In vitro, SXN markedly enhanced the proliferation of bone marrow nucleated cell (BMNC) and the colony-forming capacity of BFU-E, CFU-E, and CFU-GM, while also elevating the expression of proteins involved in the SCF/c-KIT/PI3K/AKT pathway in BMNC. Additionally, SXN enhanced the proliferation and differentiation of mesenchymal stem cell (MSC) and increased SCF secretion. In conclusion, SXN demonstrates the capacity to enhance erythropoiesis by upregulating SCF expression, thereby promoting HSPC proliferation and differentiation via the SCF/c-KIT/PI3K/AKT pathway. SXN may offer a new strategy for improving the activity of HSPC and promoting erythropoiesis in the treatment of hematopoiesis disorders.

## 1. Introduction

Erythropoiesis is a crucial physiological process that ensures the maintenance of hemoglobin homeostasis and facilitates effective oxygen transport. The initial phases of erythropoiesis are derived from multipotential hematopoietic stem and progenitor cell (HSPC). In response to a range of cytokines, HSPC undergo differentiation into erythroid-committed progenitor cell, which are typically identified as burst-forming unit erythroid (BFU-E) and subsequently differentiated colony-forming unit erythroid (CFU-E) [1]. HSPC are essential for regenerating the entire hematopoietic system, demonstrating remarkable long-term self-renewal and multidirectional differentiation capacities [2,3]. They play a central role in both steady-state and stress-induced hematopoiesis [4]. The hematopoietic niche, also referred to as the microenvironment, is a crucial regulator of HSPC proliferation, self-renewal, and differentiation [5,6,7]. This niche comprises stromal cells and various cytokines. Stromal cells provide structural support, while cytokines regulate the maintenance, migration, proliferation, differentiation, and maturation of HSPC under both steady-state conditions and in response to hematopoietic injury [8,9]. Among these cytokines, stem cell factor (SCF) interacts with the KIT receptor to enhance the proliferation and survival of HSPC, BFU-E, CFU-E, and proerythroblasts [10,11]. SCF is primarily secreted by stromal cells in the niche, including adipocytes, fibroblasts, and endothelial cells, which originate from mesenchymal stem cell (MSC). SCF promotes the proliferation and differentiation of HSPC by interacting with the receptor c-KIT through the PI3K/AKT signaling pathway. The secretion of SCF from rapidly proliferating adipocytes facilitates hematopoietic recovery in the bone marrow of mice following X-ray radiation exposure [12]. Thus, SCF is indispensable for both the regeneration of HSPC and stress-induced hematopoiesis [13,14]. Recombinant human stem cell factor (rhSCF) is commonly used for the ex vivo amplification of HSC, but its receptors are also present on mast cells and cancer cells, limiting its direct clinical use in vivo [15,16]. Therefore, stimulating the autocrine secretion of SCF in the hematopoietic microenvironment to promote the proliferation of HSPC may be a more effective strategy to promote erythropoiesis.

Sheng Xue Ning (SXN), mainly composed of sodium iron chlorophyllin (Fe-chlorine iron p6, Fe-chlorine e6, Fe-isochlorin e4, etc.), are refined through the dissolution, saponification, extraction, acid precipitation, washing, and substitution of the magnesium ion in the chlorophyll center with an iron ion [17]. Studies have shown that chlorophyll can promote the proliferation of hematopoietic progenitor cell [18]. Given that the structure of chlorophyll is similar to heme, replacing the magnesium ions in the porphyrin ring with iron ions may theoretically enhance its pharmacological role in iron supplementation and erythropoiesis improvement [19,20]. Clinical evidence indicates that SXN is effective in treating iron deficiency anemia, renal anemia, and anemia during pregnancy with minimal side effects [21,22]. Research reports that SXN can effectively stimulate red blood cell production by promoting erythropoietin (EPO) synthesis and regulating iron homeostasis in adenine-induced anemia [23]. However, the specific effects of SXN on hematopoietic cell lineages, particularly its molecular mechanism in enhancing HSPC function, remain to be fully understood. Preliminary studies suggest that SXN increases nucleated cell counts in bone marrow, elevates the proportion of c-KIT^+^ HSPC, and boosts SCF mRNA expression in cyclophosphamide-induced anemia mice [24]. These findings lead us to hypothesize that SXN may enhance SCF secretion, thereby promoting the proliferation and differentiation of HSPC.

This study utilized a total-body X-ray irradiation mouse model to investigate the effects of SXN on blood cell count and the proliferation and differentiation of HSPC. Our results confirm for the first time that SXN can promote MSC proliferation and differentiation and upregulate SCF expression, thereby enhancing HSPC proliferation and promoting erythropoiesis. This research may offer a novel therapeutic strategy to accelerate hematopoietic recovery by promoting HSPC proliferation and differentiation.

## 2. Materials and Methods

### 2.1. Reagents and Antibodies

SXN (Lot: 20190306) was provided by Wuhan United Pharmaceutical Co., Ltd. and stored at 4 °C. FITC anti-CD3(E-AB-F1013C), FITC anti-Gr-1(E-AB-F1120C), FITC anti-CD11b(E-AB-F1081C), FITC anti-CD45R(E-AB-F1112C), and FITC anti-Ter119 (E-AB-F1125C); PE anti-Sca-1(E-AB-F1191D); APC anti-c-KIT(E-AB-F1092E); PE-Cy7 anti-CD127 (E-AB-F1023E); PE/CY5.5 Rat Anti-Mouse 7AAD(E-CK-A162); and APC-Cy7 anti-CD34(E-AB-F1284E) were purchased from Elabscience (Wuhan, China). Mouse bone marrow mesenchymal stem cells (MUBMX-01001) and OriCell Basal Medium (MUBMX-90011) were purchased from Cyagen Biosciences (Guangzhou, China). Antibodies against c-KIT (E-AB-70340), SCF (26582-1-AP), β-actin (E-AB-40338), and GAPDH (E-AB-40337) were purchased from Elabscience (Wuhan, China). Antibodies against P-PI3K (T40064S), and PI3K (T40064S) were purchased from Abmart (Shanghai, China). Antibodies against P-AKT (80455-1-RR), and AKT (10176-2-AP) were purchased from Proteintech (Shanghai, China). A CCK-8 assay was purchased from APExBIO (Houston, CA, USA). FICOLL PAQUE PLUS (10327061) was purchased from Cytiva (Wales, UK).

### 2.2. Animals

Kunming (KM) mice were obtained from the Chengdu Dossy Experimental Animal CO., LTD (Chengdu, China). The mice were raised in sterile laminar flow cabinets under a 12- h light/dark cycle, and they were given standard diets and unlimited access to water. The ambient temperature was maintained at (24 ± 2) °C. Following a one-week acclimation period, the mice were assigned randomly to 5 different groups, including the control group, X-ray irradiation (model) group, X-ray irradiation + EPO (positive control) group, and X-ray irradiation + SXN (treatment) groups. Groups treated with 78 mg/kg and 156 mg/kg SXN were referred to as SXN-L (low dose) and SXN-H (high dose), respectively. Except for the control group, a single dose of X-ray (4 Gy) was administered to all mice, aiming to create a mouse model with impaired hematopoiesis. Experimental procedures involving animals were conducted according to the guidelines set by the Committee on Animal Use and Care of Southwest Medical University (Permission NO. 2020309) in Luzhou, China.

### 2.3. Hemanalysis

As previously reported [25], blood samples of 40 µL were initially collected from the fundus vein plexus of each mouse, and subsequently after 0, 4, 7, and 10 days. These samples were immediately mixed with a diluent of 160 µL for subsequent analysis of blood cell composition. The peripheral blood count was recorded using Sysmex XT-1800i (Kobe, Japan).

### 2.4. Flow Cytometry Analysis

Mice were sacrificed on the 7th day. Three mice were randomly chosen from each group, and their femurs and spleens were separated. The femurs were rinsed twice with 1 mL of saline to collect mouse bone marrow cells, which were then filtered through a nylon mesh (Solarbio, Beijing, China). From each sample, 200 µL of liquid containing one million cells was extracted. For antibody staining, the cells were incubated on ice for 20 min with various combinations of antibodies. BMNC were cultured in 6-well plates and treated with different concentrations of SXN (2.5, 5, 10 μg/mL) for 6 days. Afterward, they were washed and re-suspended with PBS. They were incubated with 5 μL of APC anti-mouse c-Kit (Elabscience, Wuhan, China) for 20 min under dark conditions. Flow cytometric analysis was performed utilizing BD FACSCanto II flow cytometry equipment (BD Biosciences, SAN Jose, CA, USA), and the results were analyzed using Flow Jo software version 10.

### 2.5. iTraq Quantitative Proteomics

iTraq quantitative protein analysis was performed by MONITOR HELIX Biotechnology Co. (Shanghai, China). After extracting the sample proteins, their concentration was determined. Based on the quantitative results, enzymatic digestion and peptide labeling were performed on the protein samples. Subsequently, the labeled peptide fractions were separated and identified using mass spectrometry. The qualitative results were obtained by submitting the raw plot files of peptide identification from Q ExactiveHF to SEQUEST software to a database search using Proteome Discoverer 1.3 (Thermo Scientific, Waltham, MA, USA) software.

### 2.6. Immunofluorescence Staining

Mouse femora were fixed overnight in 4% paraformaldehyde, followed by decalcification using a decalcifying agent. The tissue was subsequently encased in paraffin and sliced into sections following submersion in paraffin. Tissue sections were subjected to antigen retrieval using EDTA repair solution (pH 8.0) in a microwave oven. Subsequently, the prepared primary antibody was carefully applied onto the sections, ensuring flat placement within a moist chamber, and incubated overnight at 4 °C. Slides were briefly immersed in PBS (pH 7.4) for 5 min with gentle agitation using a shaker for decolorization purposes. Corresponding secondary antibodies were applied and incubated at room temperature under dark conditions. DAPI solution was added into designated areas on the slides and incubated at room temperature under dark conditions as well. Autofluorescence quenching agent B solution was subsequently added and rinsed with running water afterwards. Finally, slides were covered with an anti-fading mounting medium and imaged using a fluorescence microscope (Nikon, Tokyo, Japan).

### 2.7. Histopathological Analysis

The femurs were immersed in 10% formaldehyde solution for a duration of 24 h, followed by decalcification using a solution for more than one month. Subsequently, the samples were embedded in paraffin and sliced into sections measuring 5 µm. These sections were then subjected to staining with hematoxylin and eosin (H&E) before being examined under an Olympus BX51 microscope (Olympus Optical, Tokyo, Japan). For each sample, relevant images were captured from three distinct perspectives.

### 2.8. Cell Culture

Cells were incubated in a humidified incubator with 5% CO_2_ at a temperature of 37 °C. MSC were cultured at densities of 2.5~4 × 10^4^ cells/cm^2^. Upon reaching approximately 80% confluency, the cells were detached using trypsin and transferred into 6-well plates for subsequent experiments. BMNC Collection Procedure: Add 3 mL FICOLL PAQUE PLUS (Cytiva, Wales, UK) along the bottom of the centrifuge tube wall to the lower layer of bone marrow cells, centrifuge at 400× *g* for 20 min, collect the middle layer cells, wash three times with serum-free medium, and resuspend in low-sugar DMEM medium containing 10% horse serum and 10% fetal bovine serum.

### 2.9. Cell Viability Assay

MSC were cultured in 96-well plates at a density of 2.0 × 10^4^ cells per well and exposed to varying concentrations of SXN (2.5, 5, and 10 µg/mL) for 3 days. The control group consisted of untreated cells. To assess the viability of the cells, we conducted the CCK-8 assay based on the guidelines provided by APExBIO (Houston, CA, USA). The absorbance at 450 nm was measured using BioTek (Winooski, VT, USA).

### 2.10. Cell Morphological Observations

To examine changes in cell morphology during the cultivation process, cells were cultured in 6-well plates with a cell density of 1 × 10^6^ cells per well [26]. After treatment with SXN (2.5, 5, and 10 µg/mL) for 3 days, MSC and BMNC from each group were visually assessed using an inverted microscope (Nikon, Tokyo, Japan), and images under bright field conditions were captured. Then, they underwent morphological observation and photography.

### 2.11. Cell Differentiation

The differentiation potential to adipocytes was confirmed by a differentiation assay. For this assay, the MSC cells were cultured with adipocyte media (Cyagen Biosciences, Guangzhou, China) for 18 days and then stained with Oil Red O to confirm the presence of adipocytes [27].

### 2.12. Colony Formation Assay

The BMNC were grown in MethoCult ™ GF M3434 semisolid medium (STEMCELL Technologies, Vancouver, Canada) with a cell density of 1 × 10^5^ cells per well. According to the “Mouse Colony-Forming Unit Assays” operating instructions, the cells were observed and counted under a microscope.

### 2.13. Western Blot

The femurs of mice were extracted, and bone marrow cells were rinsed with saline solution, followed by lysing at a temperature of −80 °C using 1 × RIPA lysis buffer (Cell Signaling Technologies, Beverly, MA, USA) for protein extraction. MSC were cultured in 6-well plates with a density of 4 × 10^4^ cells/well and treated with SXN (2.5, 5, and 10 µg/mL) for a duration of 3 days. Afterward, the culture medium was discarded and proteins were extracted according to the above method. Protein concentrations were determined utilizing the BCA Protein Assay Kit (EpiZyme, Shanghai, China). SDS-PAGE was utilized for protein separation, followed by a transfer onto a PVDF membrane. Subsequently, the membranes were blocked in a solution comprising 0.1% PBS–Tween and 10% nonfat desiccated milk prior to being exposed to primary antibodies for a duration of 8 h. Subsequently, the membranes underwent rinsing with PBST (PBS containing 2% Tween-20), followed by the addition of secondary antibodies for 1 h. The ChemiDoc MP Imaging System (Bio-Rad, Hercules, CA, USA) was employed for protein band detection after visualization using ECL Western blotting detection reagent (4A Biotech Co., Ltd., Beijing, China). ImageJ software was utilized to quantify the gray value of the protein bands. The relative image intensity of the target protein in comparison to GAPDH or β-actin is indicative of their expression.

### 2.14. Statistical Analysis

The statistical analysis was carried out using GraphPad Prism 9.0 (GraphPad Software, La Jolla, CA, USA), and the data were presented as mean ± standard deviations. All experiments were repeated three times. To determine the significance between two groups, Student’s *t*-test was utilized, whereas for comparisons involving three or more groups, one-way analysis of variance was employed. Statistical significance was considered at *p* < 0.05.

## 3. Results

### 3.1. SXN Facilitated the Recovery of the Peripheral Blood Cell Count in X-ray Irradiated Mice

To evaluate the effect of SXN on erythropoiesis following X-ray irradiation, KM mice were exposed to 4 Gy X-rays and subsequently treated with SXN [28]. The treatment regimen involved oral administration of SXN at doses of 156 and 78 mg/kg daily for 10 days following irradiation (Figure 1A). A significant reduction in white blood cell (WBC) levels 24 h after irradiation confirmed the successful establishment of the irradiated mouse model. Mice were then grouped based on their WBC counts for subsequent treatment with SXN or EPO (as a positive control). Notably, on the 4th, 7th, and 10th days after irradiation, marked differences in red blood cell (RBC) levels were observed between the control and model groups. The SXN treatment group demonstrated a substantial recovery in peripheral blood RBC and hemoglobin (HGB) levels (Figure 1B,C) on the 7th and 10th days. Platelet (PLT) levels across all irradiated groups reached a nadir on the 7th day post -irradiation. On the 10th day, the SXN-treated groups exhibited a significant elevation in platelet counts compared to the model group (Figure 1D). WBC levels remained consistent across all irradiated groups at each time point (Figure 1E).

Radiation-induced hematopoietic dysfunction is often attributed to the generation of reactive oxygen species (ROS) [29], leading to the depletion of rapidly proliferating bone marrow cells and severe hematopoietic system damage [30,31,32]. Flow cytometry analysis of bone marrow cells revealed elevated ROS levels in the model group compared to the normal group, with a significant reduction observed in the SXN-treated group (Figure 1F,G). H&E staining indicated significant bone marrow cell loss and structural damage in the model group, with an increased number of adipocytes and large cavity areas. In contrast, both the EPO and SXN treatment groups showed alleviated pathological changes, characterized by a higher number of nucleated cell and a proliferation of adipocytes within the bone marrow (Figure 1H). Perilipin-1 is a significant protein located on the surface of lipid droplets, predominantly expressed in adipocytes. The Western blot analysis revealed a significant increase in perilipin-1 expression in bone marrow following SXN treatment, as compared to the model group (Figure 1I). These findings suggest that SXN may facilitate the regeneration of bone marrow cells, support the formation of adipocytes, and improve hematopoietic recovery subsequent to radiation-induced injury.

### 3.2. SXN Enhanced the Proliferation and Differentiation of HSPC in X-ray Irradiated Mice

The bone marrow is the primary hematopoietic site in adults and supports the differentiation of hematopoietic stem cells into various hematopoietic progenitor cell [33]. Utilizing flow cytometry, we identified key hematopoietic stem and progenitor cell populations including HSPC (KLS, Lin^−^/Sca-1^+^/c-KIT^+^), common lymphoid progenitors (CLP, Lin^−^/CD127^+^/Sca-1^+^/c-KIT^+^), common myeloid progenitors (CMP, Lin^−^/CD127^−^/Sca-1^−^/c-KIT^+^/CD34^+^), and megakaryocyte–erythroid progenitors (MEP, Lin^−^/CD127^−^/Sca-1^−^/c-KIT^+^/CD34^−^) [34,35,36,37]. These cell populations were useful for assessing the effects of SXN on hematopoietic stem and progenitor cell after X-ray irradiation. On the 7th day after irradiation, flow cytometric analysis revealed a significant reduction in the KLS population in the bone marrow of the model group, with a notable increase in the SXN-treated group. Moreover, an increase in the populations of downstream CLP, CMP, and MEP cells was observed in the SXN group compared to the model group. These results indicate that SXN enhanced the proliferation and differentiation of HSPC in the bone marrow of irradiated mice (Figure 2A–E). Given the spleen’s role in emergency hematopoiesis [38,39,40], we also assessed the effects of SXN on splenic HSPC in irradiated mice. The results indicated a significant increase in KLS and CLP populations in the spleen of SXN-treated irradiated mice (Appendix A).

Additionally, tSNE dimensionality reduction analysis was employed to dissect the interrelationships between different hematopoietic cell populations (Figure 3A). Compared to the control group, the model group exhibited a loss of lineage^+^ mature hematopoietic cells, which reappeared in the EPO and SXN groups. Notably, the SXN group showed a more abundant lineage^+^ cell population than the EPO group. The results indicated a significant proliferation of mature hematopoietic cells after EPO and SXN treatment in X-ray irradiated mice. Compared to the model group, the overlap between the KLS and CMP or MEP cell populations increased in the SXN group. In the area between CMP and MEP, an additional KLS population appeared in the SXN group, and the analysis showed that these cells exhibited the characteristics of the CLP population. These results demonstrated that SXN administration significantly promotes the differentiation of KLS cells into CMP, MEP, and CLP populations. The EPO group only showed an increase in the overlap between the KLS and MEP populations, indicating that EPO promoted the differentiation of KLS cells into the MEP population.

Given the iron content of SXN, its primary action appears to be on terminal erythropoiesis and hemoglobin synthesis [41]. We utilized flow cytometry to examine the levels of advanced erythroid progenitor cell (CD71^+^/Ter119^+^ cell) in the bone marrow and spleen using flow cytometry. We observed that the increased levels of erythroid progenitor CD71^+^/Ter119^+^ cell in both the EPO and SXN groups compared to the model group (Figure 3B–D). The experimental results showed that SXN promotes the proliferation and further differentiation of immature HSPC to establish long-term hematopoietic function. Additionally, SXN directly affects mature hematopoietic cells, such as erythroid progenitor cell, to promote the early recovery of hematopoietic function.

### 3.3. Proteomic Analysis of the Mechanism of SXN in Enhancing HSPC Proliferation and Differentiation after X-ray Irradiation

To decipher the molecular mechanisms of SXN in hematopoietic function reconstruction, we employed iTRAQ/TMT quantitative proteomics to examine alterations in protein levels within mouse bone marrow cells post -SXN treatment. Our comprehensive analysis identified a total of 5386 proteins, with 5266 quantified. Compared to the model groups, considering a fold change of 1.2 and a significance threshold of *p* < 0.05, 291 up-regulated and 132 down-regulated proteins were revealed in the SXN-treated group (Figure 4A). The up-regulated proteins in the SXN group predominantly participated in molecular functions such as binding, catalytic activity, transporter activity, and transcription factor activity. At the cellular component level, these proteins were chiefly associated with organelles and membranes. Functional classification highlighted their involvement in various cellular processes, signaling pathways, metabolic and developmental processes, and biological adhesion (Figure 4B and Appendix A). Moreover, molecular function enrichment analysis showed that the proteins that were up-regulated in the SXN group were significantly involved in growth factor binding, macromolecular complex binding, special structure DNA binding, and transcription factor binding (Figure 4C and Appendix A). The findings suggest that SXN administration leads to a significant increase in the proliferation of bone marrow cells, potentially due to the direct impact of upregulated growth factors on promoting both the multiplication and specialization of HSPC.

### 3.4. SXN Enhanced HSPC Proliferation via the Expression of SCF and the Activation of the c-KIT PI3K/Akt Signaling Pathway

Based on previous experiments, it is speculated that SCF may contribute to the recovery of hematopoietic function in X-ray irradiated mice following SXN treatment. We focused on evaluating the expression of SCF in femoral bone marrow. Immunofluorescence staining of the femoral bone marrow revealed a marked elevation in SCF expression in the SXN groups compared to the model group (Figure 5A,B). Western blot analysis provided supporting evidence demonstrating an elevation in SCF expression in the SXN group compared to the model group (Figure 5C). This finding indicated that SXN upregulated SCF expression in the bone marrow. SCF is known to bind to its receptor c-KIT, initiating the activation of the phosphatidylinositol-3-kinase (PI3K)/AKT signaling pathway, which plays a crucial role in supporting the survival, proliferation, and differentiation processes of HSPC [42]. Meanwhile, c-KIT is a surface-specific marker of HSPC in hematopoietic tissues. Our Western blot results demonstrated that SXN treatment significantly upregulated the expression of c-KIT and the phosphorylation of PI3K and AKT in bone marrow, compared to the model group (Figure 5D–F). The same trend was also observed in the Western blot analysis of mouse splenic cells (Figure 5G–J). Therefore, we consider that SXN could increase SCF expression in hematopoietic microenvironments and activate the c-KIT/PI3K/AKT pathway, thereby promoting the proliferation of HSPC in X-ray irradiated mice.

### 3.5. SXN Enhanced the Proliferation and Differentiation of HSPC In Vitro

Based on the in vivo experimental data, we then validated the effect of SXN in vitro. We extracted mouse bone marrow nucleated cell (BMNC), and then treated them with varying SXN concentrations (2.5, 5, 10 µg/mL). The findings indicated a time-dependent increase in the quantity of BMNC within the SXN treatment group, with SXN (10 µg/mL) demonstrating the most pronounced impact. The BMNC grew in compact clusters, and on the 6th day, the cells in the SXN (10, 5 µg/mL) group were partially adherent to the wall. These cells may be MSC, suggesting that SXN may promote the proliferation of MSC (Figure 6A). To investigate the impact of SXN on the growth of BMNC, we isolated BMNC from EGFP transgenic mice and assessed the influence of SXN by analyzing the fluorescence intensity of BMNC following treatment with varying concentrations of SXN. The results showed that SXN can significantly promote the proliferation of BMNC (Figure 6B–D). Flow cytometry was used to identify c-KIT^+^HSPC in BMNC, and the results showed that SXN promoted the proliferation of HSPC in vitro compared with the control group (Figure 6E,F).

A colony-inducing culture of mouse BMNC and spleen cells was conducted, and the formation of erythroid progenitor cell colonies was observed following treatment with varying concentrations of SXN (10, 5, 2.5 µg/mL). The research results indicate that the number of BMNC and spleen cells significantly increased after SXN treatment. In BMNC, SXN demonstrated a concentration-dependent enhancement in the promotion of colony-forming unit-erythroid (CFU-E) and colony-forming unit-granulocyte and macrophage (CFU-GM), as well as colony-burst forming unit-erythroid (BFU-E). Specifically, the greatest impact was observed with SXN at a concentration of 10 µg/mL (Figure 6G). In spleen cells, the same trend of promoting colony formation was shown as in BMNC (Figure 6H). The results demonstrated that SXN significantly promoted the colony formation of erythroid progenitor cell in BMNC and spleen cell, further proving that SXN promoted the proliferation and differentiation of erythroid-committed progenitor cell in bone marrow and spleen. Western blot results showed that the expression of SCF and c-KIT and the phosphorylation of PI3K and AKT were significantly upregulated in BMNC treated with 2.5, 5, and 10 μg/mL SXN (Figure 6I–L). The results showed that SXN upregulated the expression of SCF in BMNC, activated the c-KIT/PI3K/AKT pathway, and promoted the proliferation and differentiation of HSPC.

### 3.6. SXN Upregulated the Expression of SCF by Promoting MSC Proliferation and Differentiation

Mesenchymal stem cell (MSC), capable of differentiating into various cell types including endothelial cells, adipocytes, and fibroblasts, are vital components of this environment and the source of SCF in hematopoietic microenvironments [43,44,45]. To investigate the effect of SXN on SCF secretion, we treated mouse bone MSC with varying SXN concentrations (1.25, 2.5, 5, 10, and 20 µg/mL) for 3 days. The treatment with SXN, particularly at 2.5, 5, and 10 µg/mL, increased the cell density (Figure 7A) and cell viability of MSC (Figure 7B) compared to the control group. Western blotting results revealed significant upregulation of p-PI3K and p-AKT in MSC treated with 2.5, 5, and 10 µg/mL SXN (Figure 7C,D). The results indicate that SXN could promote the proliferation of MSC. Additionally, we investigated the effect of SXN on the differentiation of MSC into adipocytes. SXN treatment at 2.5, 5, and 10 µg/mL notably increased the area of lipid droplets and perilipin-1 expression compared to the control group (Figure 7F–H), indicating that SXN significantly enhances MSC’ differentiation ability. Further analysis showed that SCF expression was substantially upregulated in MSC and adipocytes derived from MSC’ differentiation (Figure 7E,I). The collective evidence suggests that SXN administration promotes the proliferation and differentiation of MSC, while markedly enhancing the secretion of SCF.

## 4. Discussion

Erythropoiesis primarily occurs in the bone marrow and spleen, and it is substantially affected by various factors, such as chronic inflammatory diseases, cancer, and cancer therapy. In particular, the damage caused by radiotherapy and chemotherapy to hematopoietic stem cell and progenitor cell in cancer patients leads to anemia, underscoring the urgent need for effective strategies to promote red blood cell generation [46,47]. During the process of erythropoiesis, multiple external factors have been identified that regulate the differentiation, proliferation, and survival of erythroid precursor cells. Currently, the main cytokine that stimulates erythropoiesis in clinical practice is erythropoietin (EPO). It is well established that EPO can stimulate the differentiation of HSPC into BFU-E while also promoting the rapid proliferation and maturation of CFU-E to responsible for the production of red blood cells. However, the long-term and continuous use of EPO carries the risk of depleting more primitive HSPC [48,49]. Therefore, there is an urgent need for new therapeutics that can effectively enhance the proliferation and differentiation of HSPC to promote erythropoiesis.

The hematopoietic niche is a dynamic environment that regulates the behavior and function of HSPC, including self-renewal, maturation, apoptosis, quiescence, and migration. The niche comprises various stromal cells, such as mesenchymal stem cell (MSC), osteoblasts, adipocytes, and endothelial cells, which modulate HSPC’ functions through the secretion of cytokines such as SCF, IL3, and G-CSF [50]. The importance of the hematopoietic microenvironment in the expansion of HSPC, bone marrow transplantation, and regenerative medicine is increasingly recognized [51]. SCF can bind to the surface-specific marker c-KIT to regulate HSPC proliferation and differentiation, demonstrating the potential application of hematopoiesis. In addition to HSPC, c-KIT is also expressed in adult tissues such as the prostate, liver, and heart. Therefore, administering exogenous SCF has been limited to achieve the effect of regulating the proliferation of HSPC to improve erythropoiesis in vivo [52,53]. Consequently, strategies that increase SCF secretion in the hematopoietic microenvironment, promoting HSPC proliferation and differentiation to promote erythropoiesis, may provide new therapeutic options for hematopoietic dysfunction.

SXN has been clinically used in China as an iron supplement for the treatment of various types of anemia. Our investigation suggests that SXN possesses the ability to promote the proliferation of HSPC, thereby facilitating the process of erythropoiesis in the bone marrow. These findings broaden the potential applications and mechanisms of SXN in the treatment of anemia. We observed that SXN treatment increased the proportion of HSPC (KLS population) and differentiated the KLS population into CLP, MEP, and CMP in the bone marrow and spleen. Additionally, SXN was demonstrated to enhance the capacity of bone marrow and spleen cells to generate BFU-E and CFU-E colonies. This suggests that SXN contributes to the proliferation and differentiation of HSPC, thereby facilitating long-term hematopoietic establishment. Proteomic analysis revealed that SXN affects biological processes and molecular functions mainly related to cell proliferation and differentiation, potentially through the upregulation of growth factors. We further demonstrated that SXN promotes SCF secretion in hematopoietic microenvironments, which influences HSPC proliferation and differentiation by binding to the HSPC’ surface characterization marker, c-KIT. Unfortunately, mice that undergo multiple punctures of the posterior venous plexus in the fundus during the experimental procedure may suffer from blood loss, which can lead to extramedullary hematopoiesis and subsequently promote the proliferation of HSPC in the spleen. We cannot confirm the priority of SXN for stimulating hematopoiesis in the bone marrow or spleen. Adult hematopoiesis occurs primarily in the bone marrow. The spleen is an important site for extramedullary hematopoiesis in mice and humans, and the conditional deletion of SCF in splenic endothelial cells does not affect bone marrow hematopoiesis [39]. Therefore, we focused our SXN studies on bone marrow hematopoiesis. Additionally, we confirmed that SXN promotes the proliferation and differentiation of bone marrow MSC, which are cells that constitute the hematopoietic microenvironment and produce SCF in the bone marrow. These results indicate that SXN enhances the proliferation and differentiation of MSC, thereby increasing SCF production in hematopoietic microenvironments and promoting HSPC proliferation and differentiation in the bone marrow.

During erythropoiesis, iron associates with transferrin (Tf–iron) and is incorporated into proerythrocytes as a fundamental component for hemoglobin synthesis via the transferrin receptor (TfR). Recent investigations have indicated that the iron demand of HSPC escalates as these cells undergo differentiation. Notably, significant apoptosis of HSPC has been observed in the bone marrow of mice with a targeted knockout of the hematopoietic stem cell TfR1 [54]. However, heme, which is an iron-containing protein that does not rely on TfR1 for cellular entry, can substitute Tf–iron to fulfill the iron requirements of HSPC during their differentiation [55]. Furthermore, our additional research demonstrated that SXN has the capacity to directly enhance the proliferation of K562 cells and their ability to form colonies. Given the structural similarity between SXN and heme, it is postulated that SXN may enter bone marrow MSC and HSPC through endocytosis, thereby improving the hematopoietic microenvironment and promoting hematopoiesis. In conclusion, beyond its function as an iron supplement in erythrocyte maturation, our study elucidated that SXN enhances the proliferation and differentiation of MSCs and promotes HSPC proliferation, thereby accelerating erythropoiesis within the bone marrow.

## 5. Conclusions

In conclusion, this study demonstrated that SXN promotes the proliferation and differentiation of MSC and upregulates the expression of SCF in the hematopoietic microenvironment. This results in the enhancement of HSPC proliferation and differentiation via the SCF/c-KIT/PI3K/AKT pathways, thereby promoting erythropoiesis (Figure 8). These findings suggest that SXN may represent a promising therapeutic approach to promote HSPC proliferation and differentiation, facilitating long-term hematopoietic reconstitution for the treatment of anemia or hematopoietic dysfunction.

## Figures and Tables

**Figure 1 biomolecules-14-01147-f001:**
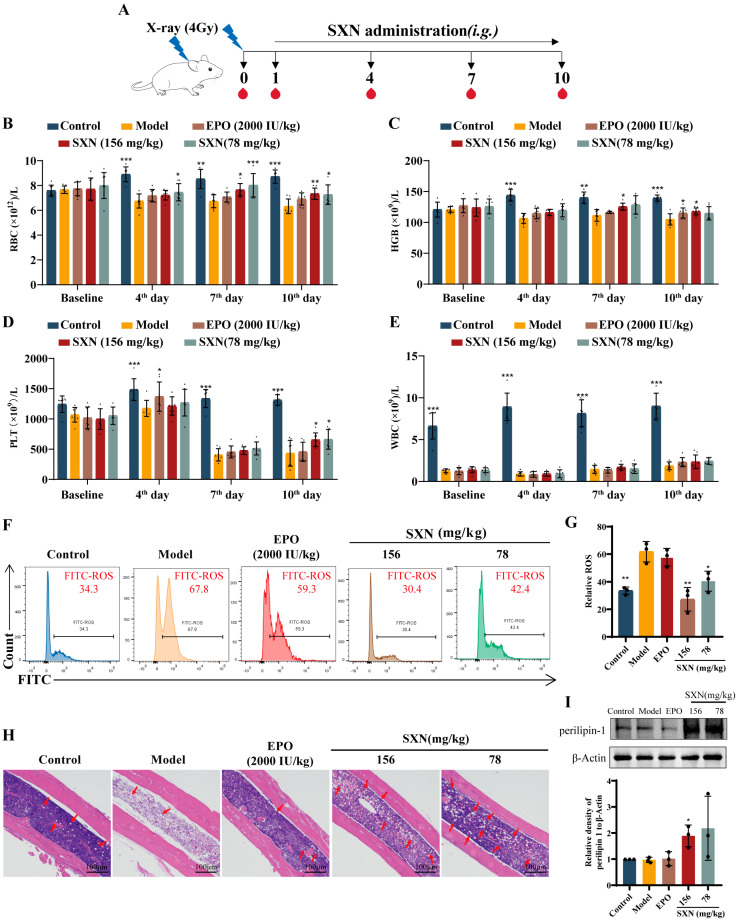
The effect of SXN on the blood cell count and bone marrow cells in X-ray irradiated mice. (**A**) Radiation exposure and administration methods in mice. Blood cell counts show (**B**) RBC, (**C**) HGB, (**D**) PLT, and (**E**) WBC on the 4th day, 7th day, and 10th day. The data are expressed as the mean ± SD. * *p* < 0.05, ** *p* < 0.01, and *** *p* < 0.001 vs. the model group; *n* = 8. (**F**,**G**) Intracellular ROS was examined and analyzed by flow cytometry in bone marrow cells, with * *p* < 0.05, ** *p* < 0.01 vs. the model group; *n* = 3. (**H**) Representative images of H&E staining of bone marrow cells (×100 magnification; scale bar: 100 µm). (**I**) Representative immunoblot images of perilipin-1 in bone marrow cells. The data are the mean ± SD (*n* = 3). * *p* < 0.05, ** *p* < 0.01 vs. the model group.

**Figure 2 biomolecules-14-01147-f002:**
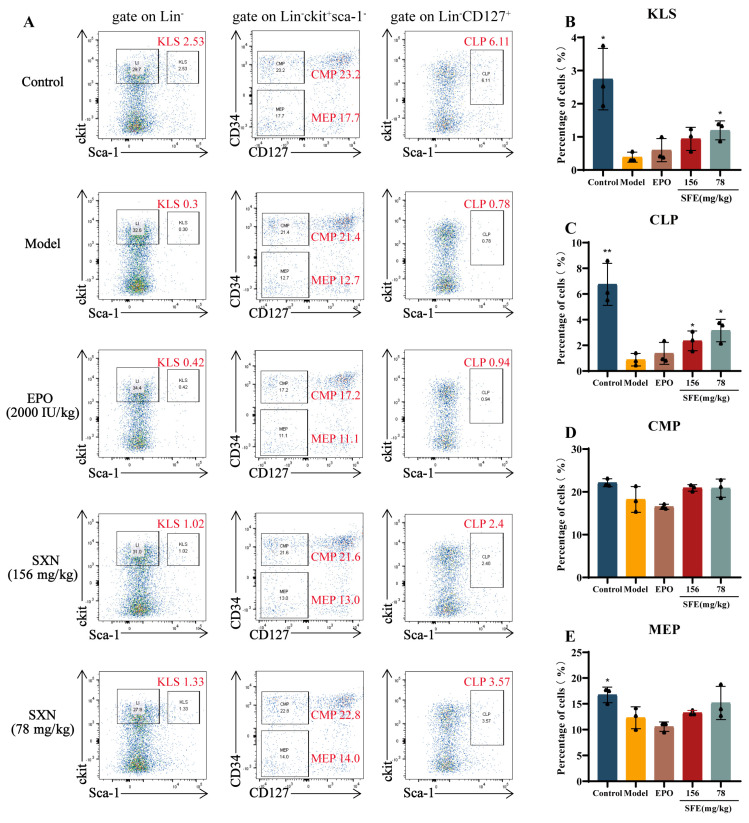
SXN promoted the proliferation of HSPC in X-ray irradiated mice in bone marrow cells. (**A**) The frequencies of HSPC (KLS, Lin^−−^/Sca-1^+^/c-KIT^+^), common lymphoid progenitor cell (CLP, Lin^−^/CD127^+^/Sca-1^+^/c-KIT^+^), common myeloid progenitor cell (CMP, Lin^−^/CD127^−^/Sca-1^−^/c-KIT^+^/CD34^+^), and megakaryocyte-erythroid progenitor cell (MEP, Lin^−^/CD127^−^/Sca-1^−^/c-KIT^+^/CD34^−^) in the bone marrow cells of each group (*n* = 3). (**B**–**E**) The histogram represents the percentage of KLS, CLP, CMP, and MEP cells in each group. The data are the mean ± SD (*n* = 3). * *p* < 0.05, ** *p* < 0.01vs. the model group.

**Figure 3 biomolecules-14-01147-f003:**
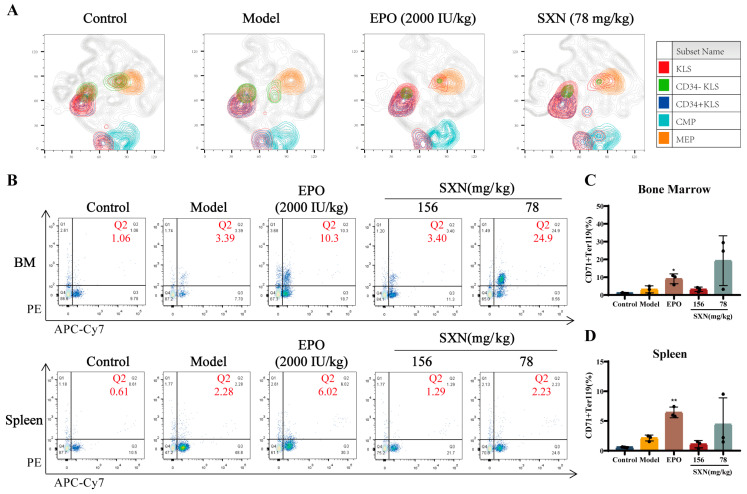
SXN promoted the differentiation of HSPC and the proliferation of erythroid progenitor cell in X-ray irradiated mice. (**A**) tSNE dimensionality reduction analysis of hematopoietic cell populations. (**B**–**D**) The frequencies of erythroid progenitor cell (Ter119^+^/CD71^+^) cell populations in the bone marrow and spleen of each group as measured by flow cytometry. The data are the mean ± SD (*n* = 3). * *p* < 0.05, ** *p* < 0.01 vs. the model group.

**Figure 4 biomolecules-14-01147-f004:**
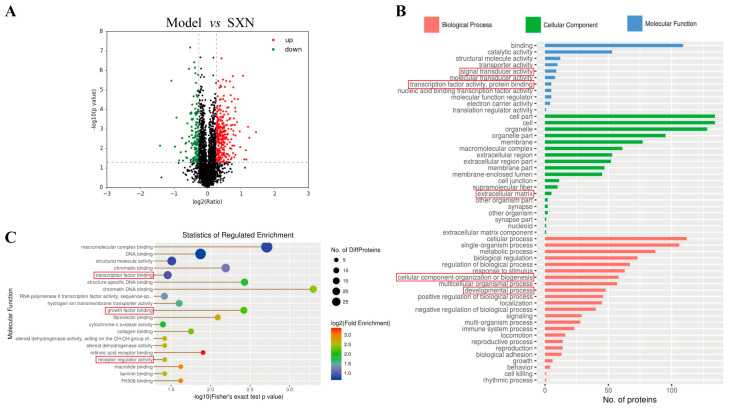
iTraq quantitative proteomics analysis. (**A**) Differentially expressed proteins between the model group and the SXN treatment group. (The red points indicate up-regulated proteins, and the green points represent down-regulated proteins). (**B**) GO enrichment analysis for the molecular function of proteins. (**C**) GO enrichment analysis and a biological process functional classification table of proteins.

**Figure 5 biomolecules-14-01147-f005:**
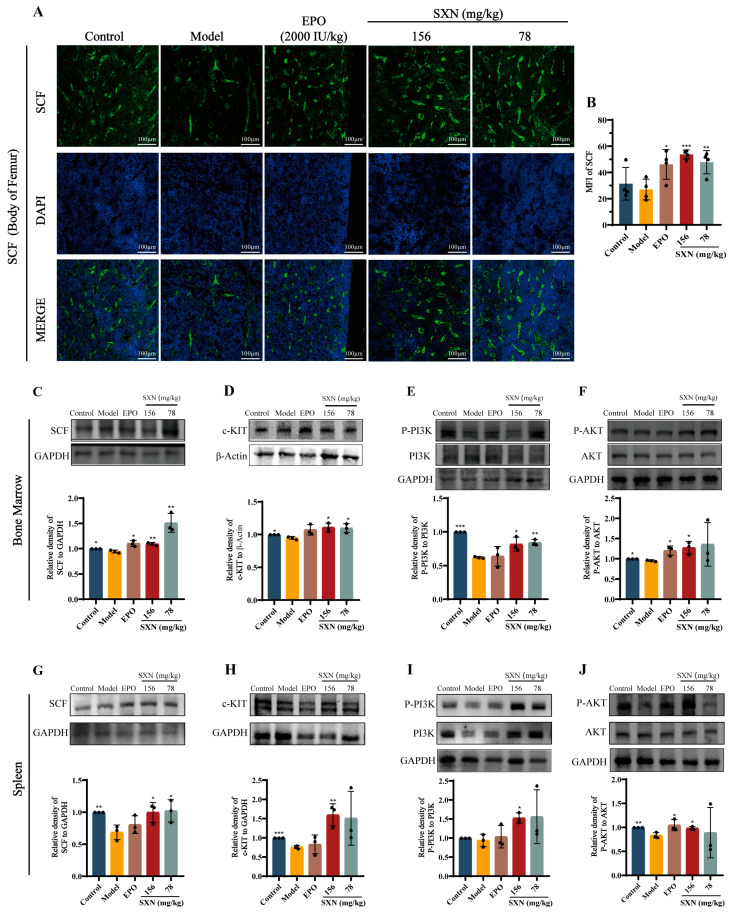
Western blot analysis of the effect of the SCF/c-KIT/PI3K/AKT signaling pathways on SXN in X-ray irradiated mice. (**A**) Immunocytochemical analysis of the expression of SCF in bone marrow cells. (**B**) Statistical results of immunofluorescence staining. Data represent the mean ± SD (*n* = 4). * *p* < 0.05, ** *p* < 0.01, and *** *p* < 0.001 vs. the model group. (**C**,**D**) Representative immunoblot images of SCF in bone marrow cells of each group mice (*n* = 3). * *p* < 0.05, ** *p* < 0.01, and *** *p* < 0.001 vs. the model group; (**E**–**G**) Representative immunoblot images of c-KIT and PI3K/Akt signaling pathways in bone marrow cells. The data are the mean ± SD (*n* = 3). * *p* < 0.05, ** *p* < 0.01, and *** *p* < 0.001 vs. the model group. (**H**–**J**) Representative immunoblot images of the SCF/c-KIT and PI3K/Akt signaling pathways in spleen cells. The data are the mean ± SD (*n* = 3). * *p* < 0.05, ** *p* < 0.01, and *** *p* < 0.001 vs. the model group.

**Figure 6 biomolecules-14-01147-f006:**
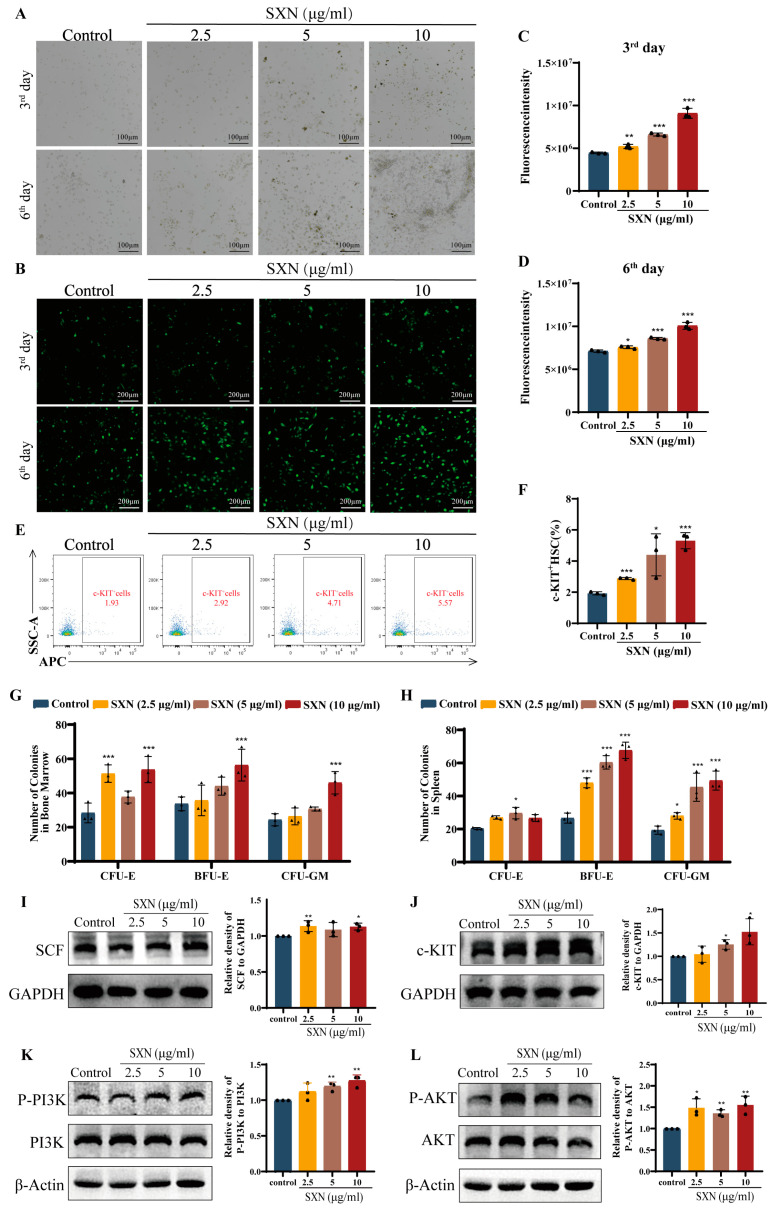
SXN enhanced the proliferation and differentiation of HSPC in vitro. (**A**) Representative images of morphological observations of BMNC on the 3rd day and 6th day. Scale bar: 100 µm. Microscopy fields were captured randomly at 10× resolution. (**B**–**D**) Representative images of morphological observations of EGFP -transgenic mouse BMNC on the 3rd day and 6th day. Scale bar: 200 µm. Microscopy fields were captured randomly at 20× resolution. (**E**,**F**) The proportion of c-KIT^+^HSPC was examined and analyzed by flow cytometry in BMNC for 6 days (*n* = 3). * *p* < 0.05, ** *p* < 0.01, and *** *p* < 0.001 vs. the control group. (**G**,**H**) The effect of SXN on the formation of CFU-E, BFU-E, and CFU-GM colonies in BMNC and spleen cells. The data are the mean ± SD (*n* = 3). * *p* < 0.05, ** *p* < 0.01, and *** *p* < 0.001 vs. the control group. (**I**–**L**) Representative immunoblot images of the SCF/c-KIT and PI3K/AKT signaling pathways after treatment with SXN (2.5, 5, and 10 µg/mL) in BMNC for 6 days. The data are the mean ± SD (*n* = 3). * *p* < 0.05, ** *p* < 0.01, and *** *p* < 0.001 vs. the control group.

**Figure 7 biomolecules-14-01147-f007:**
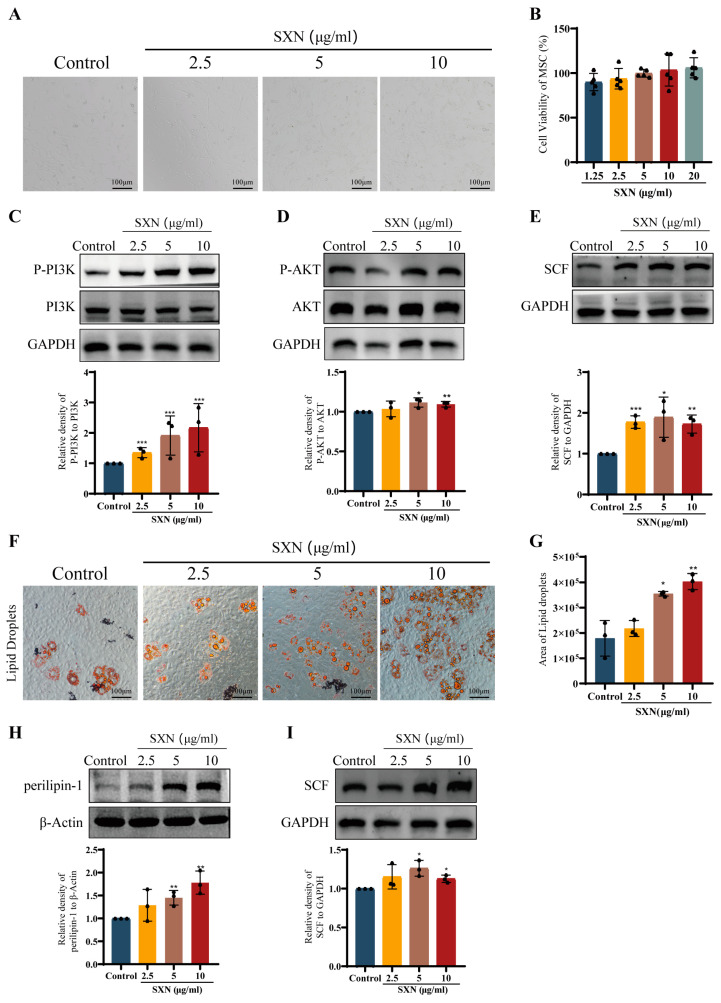
SXN upregulated the expression of SCF by promoting the proliferation and differentiation of MSC. (**A**) Representative images of morphological observations of MSC. Scale bar: 100 µm. Microscopy fields were captured randomly at 10× resolution. (**B**) Results of the CCK-8 assay for MSC proliferation. Cells were treated with different concentrations of SXN (1.25, 2.5, 5, 10 and 20 µg/mL) for 3 days. (**C**–**E**) Representative immunoblot images of the PI3K/AKT and SCF signaling pathways after treatment with SXN (2.5, 5, and 10 µg/mL) in MSC for 3 days. The data are the mean ± SD (*n* = 3). * *p* < 0.05, ** *p* < 0.01, and *** *p* < 0.001 vs. the control group. (**F**) MSC were induced to differentiate into adipocytes for 18 days. (**G**) The area of lipid droplets after adipocyte differentiation in the SXN-treated and control groups. (**H**) Representative immunoblot images of perilipin-1 in adipocytes differentiated MSC treated with SXN (2.5, 5, and 10 µg/mL). The data are the mean ± SD (*n* = 3). * *p* < 0.05, ** *p* < 0.01, and *** *p* < 0.001 vs. the control group. (**I**) Representative immunoblot images of SCF in adipocytes from differentiated MSC treated with SXN (2.5, 5, and 10 µg/mL). The data are the mean ± SD (*n* = 3). * *p* < 0.05, ** *p* < 0.01, and *** *p* < 0.001 vs. the control group.

**Figure 8 biomolecules-14-01147-f008:**
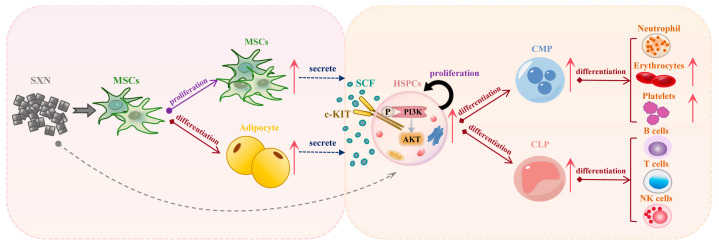
A schematic model illustrates the regulatory action of SXN on the proliferation and differentiation of mesenchymal stem cells, as well as bone marrow HSPC proliferation. SXN stimulates the secretion of stem cell factor (SCF) and upregulates the SCF/c-KIT and PI3K/AKT pathways, thereby promoting the proliferation and differentiation of in bone marrow HSPC.

## Data Availability

All data are available from the corresponding author upon request.

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
