# Peer review of "Sheng Xue Ning as a Novel Agent that Promotes SCF-Driven Hematopoietic Stem/Progenitor Cell Proliferation to Promote Erythropoiesis"

_biomolecules, 2024, doi:10.3390/biom14091147_

Round 1

Reviewer 1 Report

Comments and Suggestions for Authors

This manuscript shows that Shengxuening tablets promotes hematopoiesis, especially erythropoiesis, in response to to stress (X-ray irradiation). The mechanism is driven by the SCF/c-Kit/PI3K/Akt pathway in hematopoietic stem/progenitor cells. The authors use functional CFU assays to support the phenotypic flow cytometry-based data. Overall, this is an interesting and solid manuscript. However, a few issues regarding data interpretation and semantics should be addressed. 

The authors should use the term HSPC (hematopoietic stem/progenitor cell) rather than HSC. The lin- Sca-1+ c-Kit+ population is very heterogenous, and most of the cells in this population are not HSCs. Furthermore, HSCs can only be identified by long-term (4 month) reconstitution assays by in vivo transplantation, which the authors do not show. CFU assays are functional assays but are largely limited to myeloid-erythroid lineages. Related to this, most of the data is consistent with predominantly promoting erythopoiesis, and the authors should focus on that rather than the broader term "hematopoiesis". 

In Figure 2, the authors should double check compensation in the KLS gate... the "arm" coming out of the c-Kit+ population should not be pointing "Northeast", it should extend to the right (East). This data looks uncompensated or undercompensated. 

Comments on the Quality of English Language

No major issues

Author Response

Comments 1: The authors should use the term HSPC (hematopoietic stem/progenitor cell) rather than HSC. The lin- Sca-1+ c-Kit+ population is very heterogenous, and most of the cells in this population are not HSCs. Furthermore, HSCs can only be identified by long-term (4 month) reconstitution assays by in vivo transplantation, which the authors do not show. CFU assays are functional assays but are largely limited to myeloid-erythroid lineages. Related to this, most of the data is consistent with predominantly promoting erythropoiesis, and the authors should focus on that rather than the broader term "hematopoiesis".

Response 1: Thank you for pointing this out. We really appreciate all your comments and suggestions! Therefore, we have substituted "HSC (hematopoietic stem cell)" with "HSPC (hematopoietic stem/progenitor cell)" and "hematopoiesis" with "erythropoiesis" in the article to improve precision. Furthermore, we emphasized the significant role of HSPC in the process of erythropoiesis within both the introduction and discussion sections, and we have updated the references accordingly. We appreciate your feedback regarding the HSC transplantation experiment. Due to its extensive duration, we plan to examine the assays in our future research.

According to the comment, we have modified this expression throughout the text according to the comment. “hematopoietic stem cells” ->” hematopoietic stem/progenitor cells”. (Page 1, title, Line 1, 2; Page 1, Abstract, Line 18; Page 1, Keywords, Line 33; Page 17, Abbreviations, Line 516 ) “HSCs” -> ” HSPCs” (Page 1, Abstract, Line 18, 21, 24,29, 30; Page 2, Introduction, Line 40, 43; Page 2, Introduction, Line 47, 50, 52, 55, 59, 63, 77, 79, 82, 84, 86, 88; Page 6, Results, Line 262, 266; Page 7, Results, Line 275, 277, 300; Page 8, Results, Line 304, 305; Page 9, Results, Line 312, 316, 335; Page 10, Results, Line 342, 354, 360; Page 11, Results, Line 373; Page 12, Results, Line 385, 386, 404; Page 14, Results, Line 406, 410; Page 16, Discussion, Line 458, 462, 464, 466, 468, 470, 472, 475, 477, 482, 484, 486, 491, 496, 500; Page 16, Conclusions, Line 502; Page 17, Conclusions, Line 502, 507, 510; Page 17, Abbreviations, Line 516.). “reestablish hematopoiesis” ->” promote erythropoiesis” (Page 1, title, Line 3,4; Page 1, Abstract, Line 18, 31; Page 2, Introduction, Line 63, 86; Page 17, Conclusions, Line 464, 478;). “hematopoiesis” ->” erythropoiesis” (Page 1, Keywords, Line 33; Page 5, Results, Line 224; Page 16, Discussion, Line 476, 482).

In view of the revised terminology, we had been proceed to provide a new description of the introduction and discussion sections.

In the Abstract, this first sentence was rephrased according to the comment. Stimulating erythropoiesis is essential in treatment of various anemia. (Page 1, Abstract, Line 15)

In the Introduction, we had redescribed the important role of HSPCs in the process of erythropoiesis. Erythropoiesis is a crucial physiological process that ensures the maintenance of hemoglobin homeostasis and facilitates effective oxygen transport. The initial phases of erythropoiesis are derived from multipotential hematopoietic stem and progenitor cells (HSPCs). In response to a range of cytokines, HSPCs undergo differentiation into erythroid-committed progenitor cells, which are typically identified as burst-forming unit erythrocytes (BFU-E) and subsequently differentiated colony-forming unit erythrocytes (CFU-E) [1]. (Page 1, Introduction, Line 37-43) [1] Alexis, L. C.; Vijay, G. S.; Molecular and cellular mechanisms that regulate human erythropoiesis. Blood 2022, 139, (16):,2450–2459.

In the discussion, we provided an updated account of the current clinical treatment status of erythropoiesis. Erythropoiesis primarily occurs in the bone marrow and spleen, and is substantially affected by various factors, such as chronic inflammatory diseases, cancer, and cancer therapy. In particular, the damage caused by radiotherapy and chemotherapy to hematopoietic stem cells and progenitor cells in cancer patients leads to anemia, underscoring the urgent need for effective strategies to promote red blood cell generation [46, 47]. During the process of erythropoiesis, multiple external factors have been identified that regulate the differentiation, proliferation, and survival of erythroid precursor cells. Currently, the main cytokine that stimulates erythropoiesis in clinical practice is erythropoietin (EPO). It is well established that EPO can stimulate the differentiation of HSPCs into BFU-E, while also promoting the rapid proliferation and maturation of CFU-E to responsible for the production of red blood cells. However, long-term and continuous use of EPO carries the risk of depleting more primitive HSPCs [48, 49]. Therefore, there is an urgent need for new therapeutics that can effectively enhance the proliferation and differentiation of HSPCs to promote erythropoiesis. (Page 16, Discussion, Line 451-464)

Comments 2: In Figure 2, the authors should double check compensation in the KLS gate... the "arm" coming out of the c-Kit+ population should not be pointing "Northeast", it should extend to the right (East). This data looks uncompensated or undercompensated.

Response 2: Thank you for pointing this out. We reanalyzed the data and replaced the relevant Figure2, Figure S2.and statistical charts.

Please find the figure in the attachment.

Figure 2. SXN promoted the proliferation of HSPCs in X-ray irradiated mice in bone marrow cells. (Page 8, Results, Line 303)

Please find the figure in the attachment.

Figure S2. SFE promoted the proliferation of HSCs in X-ray irradiated mice in spleen. (Supplementary data, Figure S2.)

Reviewer 2 Report

Comments and Suggestions for Authors

Shengxuening tablets (SXN) are used in China to treat iron deficiency anemia, renal anemia, and anemia in pregnancy. This research investigates SXN's role in promoting hematopoietic stem cell (HSC) proliferation to restore hematopoiesis. Using an X-ray-induced hematopoietic dysfunction model, SXN significantly promoted the recovery of peripheral erythroid cells and enhanced the proliferation and differentiation of Lin-/c-KIT+/Sca-1+ HSCs. SXN also enhanced the proliferation of bone marrow nucleated cells and mesenchymal stem cells and increased SCF secretion. The study suggests SXN could be a novel approach for enhancing HSC activity and improving hematopoietic function in treating hematopoietic disorders. However, certain matters need to be resolved. 

1.     The tablets should be corrected to match the brand name, Sheng Xue Ning (SXN).      

2.     The animal study was meticulously planned and carried out with exceptional creativity. However, the authors need to include the actual molecular structure, rather than just mentioning the molecular derivatives of chlorophyll.

3.     The authors indicated that SXN tablets are frequently employed in China for the treatment of iron deficiency anemia, renal anemia, and anemia during pregnancy. Currently, it is not customary to use animal studies as evidence to demonstrate the proliferation of hematopoietic stem cells for the purpose of restoring hematopoiesis. Anemia studies are always conducted prior to human clinical trials.

Comments on the Quality of English Language

The English was of average quality.

Author Response

Comments 1: The tablets should be corrected to match the brand name, Sheng Xue Ning (SXN).

Response 1: Thank you for pointing this out. We really appreciate all your comments and suggestions! We have been substituted "Shengxuening tablets" with "Sheng Xue Ning" in the article.

 “Shengxuening tablets” ->” Sheng Xue Ning”. (Page 1, title, Line 2; Page 1, Abstract, Line 15; Page 1, Keywords, Line 33; Page 2, Introduction, Line 65; Page 16, Discussion, Line 480; Page 17, Abbreviations, Line 516.)

Comments 2: The animal study was meticulously planned and carried out with exceptional creativity. However, the authors need to include the actual molecular structure, rather than just mentioning the molecular derivatives of chlorophyll.

Response 2: Thank you very much for your helpful suggestion and guidance. According to the references “study on fingerprint of shengxuening tablets, we have redescribed in the article that the principal component of SXN is sodium iron chlorophyllin, and its structural formula primarily includes Fe-chlorine iron p6, Fe-chlorine e6, Fe-isochlorin e4.

According to the comment, we have been redescribed the molecular structure of Sheng Xue Ning. Sheng Xue Ning (SXN), mainly composed of sodium iron chlorophyllin (Fe-chlorine iron p6, Fe-chlorine e6, Fe-isochlorin e4), are refined through dissolution, saponification, extraction, acid precipitation, washing, and substitution of the magnesium ion in the chlorophyll center with an iron ion [17]. (Page 2, Introduction, Line 65, 66.)

[17] Nie, J.; Hu, H. S.; Chen, X. Y.; Liu, J. J.; Li, K.; Luo, J.; Study on fingerprint of shengxuening tablets. Zhongguo Zhong yao za zhi 2013, 38, (20), 3502–3506.

Comments 3:  The authors indicated that SXN tablets are frequently employed in China for the treatment of iron deficiency anemia, renal anemia, and anemia during pregnancy. Currently, it is not customary to use animal studies as evidence to demonstrate the proliferation of hematopoietic stem cells for the purpose of restoring hematopoiesis. Anemia studies are always conducted prior to human clinical trials.

Response 3: Thank you for your rigorous thinking and constructive suggestion. SXN is commonly used in clinical practice to treat various types of anemia as an iron supplementation machine. Prior to drug registration, SXN had already completed research on promoting erythropoiesis in animals with hemorrhagic anemia or iron deficiency anemia. While investigating the mechanism of SXN, we accidental discovered that it stimulates SCF production, and stimulate c-KIT+ hematopoietic cell proliferation. Therefore, we designed relevant animal experiments to verify that SXN promotes hematopoiesis through the SCF/CKIT/PI3K/AKT pathway, in addition to iron supplementation.
